# Hydrophobic Modification of Biopolymer Aerogels by Cold Plasma Coating

**DOI:** 10.3390/polym13173000

**Published:** 2021-09-04

**Authors:** Baldur Schroeter, Isabella Jung, Katharina Bauer, Pavel Gurikov, Irina Smirnova

**Affiliations:** 1Institute for Thermal Separation Processes, Hamburg University of Technology, Eißendorfer Straße 38, 21073 Hamburg, Germany; Isabella.Jung@tuhh.de (I.J.); Irina.Smirnova@tuhh.de (I.S.); 2Diener Electronic GmbH & Co. KG, Nagolder Straße 61, 72224 Ebhausen, Germany; Katharina-bauer@mein.gmx; 3Laboratory for Development and Modelling of Novel Nanoporous Materials, Hamburg University of Technology, Eißendorfer Straße 38, 21073 Hamburg, Germany; Pavel.Gurikov@tuhh.de

**Keywords:** aerogels, biopolymers, cold plasma coating, hydrophobization, pore structure

## Abstract

The aim of this work was to evaluate the potential of cold plasma polymerization as a simple, fast and versatile technique for deposition of protective hydrophobic and oleophobic polymer layers on hydrophilic biopolymer aerogels. Polymerization of different fluorinated monomers (octafluorocyclobutane C_4_F_8_ and perfluoro-acrylates PFAC-6 and PFAC-8) on aerogel monoliths derived from alginate, cellulose, whey protein isolate (WPI) and potato protein isolate (PPI) resulted in fast and significant surface hydrophobization after short process times of 5 min and led to superhydrophobic surfaces with static water contact angles up to 154° after application of poly-C_4_F_8_ coatings. Simultaneous introduction of hydro- and oleophobicity was possible by deposition of perfluoro-acrylates. While the porous structure of aerogels stayed intact during the process, polymerization inside the aerogels pores led to the generation of new porous moieties and resulted therefore in significant increase in the specific surface area. The magnitude of the effect depended on the individual process settings and on the overall porosity of the substrates. A maximization of specific surface area increase (+179 m^2^/g) was obtained by applying a pulsed wave mode in the C_4_F_8_-coating of alginate aerogels.

## 1. Introduction

Aerogels are solid materials with high porosity, low density and high specific surface areas, which can be produced from a variety of starting materials such as silica, synthetic polymers and biopolymers. Due to sustainability aspects as well as low material costs, high availability, biocompatibility and non-toxicity of the educts, research on biopolymer-based aerogels has seen significantly increased importance in the last decade [1,2]. The typical steps of biopolymer–aerogel synthesis involve the formation of a hydrogel in an aqueous phase, followed by a solvent exchange and subsequent supercritical drying or, in special cases, freeze drying [3,4]. Most biopolymer-based aerogels are derived from polysaccharides, common examples being alginate and cellulose aerogels. They contain low initial biopolymer content of approximately 1.0–7.0 wt%, demonstrate high porosities (ε) up to 99%, low envelope densities (ρ_e_) of approximately 0.07–0.5 g/cm^3^ and specific surface areas in the typical range from 200–600 m^2^/g [4]. Recent studies showed that aerogels with considerable surface areas of approximately 300–400 m^2^/g and significantly higher biopolymer content of 10–20 wt% can be obtained by a controlled gelation of proteins such as whey protein isolate (WPI) and potato protein isolate (PPI) [5,6]. Since all biopolymer aerogels are characterized by a particularly high surface concentration and a variety of functionalities on the surface, such as OH-, carboxy-, amine- and amide groups, they show high potential in adsorption and separation processes and offer the possibility of targeted post-functionalization. Due to the combination of tailorable microstructures and high porosities, further application areas are as high performance thermal insulation materials and carrier matrices for targeted drug delivery [7,8]. Although, from a material standpoint, aerogels offer high variety and potential, the use of aerogels poses challenges in handling and storage: high polarity (in case of polysaccharide gels) and open-porous structure (for all aerogel types) makes biopolymer aerogels especially susceptible to moisture that can penetrate into the pores, resulting in changes of the internal surface and pore collapse. Furthermore, contact of biopolymer aerogels, e.g., based on alginate and cellulose, with polar liquids may lead to destruction of the 3D structure, and in the case of non-polar liquids, may lead to uptake of the liquid into the pores. Since surface sensitivity, storage and aging behavior are crucial properties, strategies for aerogel protection are of high relevance.

Hydrophobic post modifications of dry biopolymer aerogels may offer protection against polar substances and can be applied principally in two ways: application of an (closed) outer layer on the aerogel’s surface or by modification of the inner porous surface, whereas the favored strategy depends on the final application. The former can be achieved via different strategies, e.g., chemical vapor deposition (CVD) of coating materials from the gas phase or spray coating of aerogel particles with polymeric solutions/dispersions [7]. The drawbacks of these methods include the limitations of specific material systems, the need for high temperatures in particular cases, the high consumption of educts (for CVD) and the possibility of pore collapse due to solvent entering the non-protected aerogels pores during the process (in spray coating process) [7,9,10,11].

A promising solvent-free alternative approach is the deposition of hydrophobic polymer layers via cold plasma coating processes, wherein dry aerogels are exposed to cold plasma generated by glow discharging of a gaseous hydrophobizing agents in moderate vacuum [7]. Plasma polymerization is a complex process, involving the simultaneous generation and reaction of short-lived species. Important process parameters that determine the fragmentation of monomers and crosslinking degree of the formed polymer films as well as the film formation kinetics are deposition time (*t*), monomer flow rate, system pressure and power input (*p*_input_) [9,12]. The power input can be controlled principally via two ways: by adjusting the glow discharge power in a continuous wave mode (CW) or by modulating the power in a pulsed wave mode (PW), whereas the chemical structure and composition of PW plasma polymer coatings depends on the duty cycle (*DC*) [9,13], which reflects the ratio of pulse-on and pulse-off times:(1)$$DC\;\left(\%\right)=\frac{t_{on}}{t_{on}+t_{off}}\cdot 100$$
where *t_on_* = pulse on time, *t_off_* = pulse off time.

The thickness of polymer layers in cold plasma polymerization depends heavily on the mass deposition rate of solid from the gas/plasma phase, which is therefore a crucial parameter for the formation of stable layers and depends on the individual monomer characteristics and operational parameters as well as on plasma chamber design [12]. In addition, a freshly deposited material is subject to fragmentation and ablation by the plasma, from which the deposition occurred [12].

Cold plasma coating processes provide intrinsic advantages, such as short processing times and efficiency in terms of educt consumption; no need for wet chemicals, solvents and purification steps (and therefore environmental pollution-free); controlled attachment of nano-meter scaled layers; high variety of usable coating materials with different functionalities; and the possibility of applying controlled and tunable wettability to different surfaces [9,14,15,16,17]. Deposition of thin polymer coatings by cold plasma technology is very flexible, allowing functionalization of practically any kind of substrate, including polymers, glass, ceramics and metals, whereas the adhesion of a plasma-polymerized film to a substrate is the most important factor for the success of the surface modification process [18]. In general, better adhesion is achieved by slower deposition rates, small film thicknesses and with polar substrates [18]. Although the above given aspects suggest that cold plasma coating is a promising tool for hydrophobic post modification of polar biopolymer aerogels, only a few examples of the combination of cold plasma hydrophobization with aerogels substrates are documented in the literature. Hydrophobization of the external surface of hydrophilic cellulose and alginate aerogel substrates obtained by freeze drying was achieved by the use of trimethylchlorosilane and tetrachloromethane monomers in short process times of approximately 3–20 min (CW mode, input power ~150–250 W) in several works, leading to protection of the surfaces against water and resulting in static water contact angles (θ) in the range of approximately 102–150° [19,20,21,22]. While the mentioned studies demonstrate the principal possibility of achieving hydrophobic to superhydrophobic surfaces on porous biopolymer substrates, it remains unclear if the modification is limited to the outer surface or if the inner porous structure is also affected. Additionally, freeze-drying was used as the drying technique in all cases, which is known to lead to significantly lower specific surface areas compared with supercritical drying [1,3,23,24]. Consequently, specific surface areas in the cited examples can be expected to be low. Furthermore, pore size is expected to be rather larger (a few micrometers) than in typical aerogels from supercritical drying (a few tens of nanometers) [23].

A promising non-toxic alternative to the use of chloromethanes and chlorosilanes is the application of fluorocarbon-based polymer coatings since they exhibit extremely low surface energies. Fluoropolymer layers satisfy the widely accepted regulatory assessment criteria to be considered as “polymers of low concern” [25]. The use of octafluorocyclobutane (C_4_F_8_) as a coating agent has been documented to result in smooth, relatively robust and hydrophobic surfaces due to the formation of highly cross-linked CF_2_ and CF_3_ groups, whereas the layer thickness, deposition rates and crosslinking degree can be controlled by the power input energy and input mode [26]. Superhydrophobic surfaces can be obtained by combination of C_4_F_8_-plasma coating and nano-texturing of the substrates [16]. Simultaneous introduction of (super)-hydrophobicity and oleophobicity is possible by using perfluoro-acrylates such as 1H,1H,2H,2H-perfluorooctyl acrylate (PFAC-6) and 1H,1H,2H,2H-Heptadecafluorodecylacrylate (PFAC-8) as precursors [9,13].

The aim of this work was to evaluate the potential of cold plasma coating processes as a simple and efficient one step-technique for post modification of biopolymer aerogels with high surface areas produced via supercritical drying. The main question to be answered was whether it is possible to achieve liquid repellency of hydrophilic and open porous aerogel surfaces using fluorocarbon-based coating materials against both polar and non-polar liquids. In addition, influences of the coating process on the inner pore structure were evaluated.

Our strategy towards plasma-modified biopolymer aerogels involved the following steps: (1) Synthesis of biopolymer aerogel monoliths with different porosities and surface energies via gelation, solvent exchange and supercritical drying. Alginate and cellulose-based aerogels were used as model systems for highly hydrophilic gels with low intrinsic biopolymer content ≤6 wt% and high porosity. PPI- and WPI-based aerogels were used as substrates with higher intrinsic biopolymer content ≥15 wt% and therefore lower porosities as well as lower surface energies and additional functionalities compared with the polysaccharides. (2) Cold plasma coating of the aerogel substrates using different fluorinated precursors (C_4_F_8_, PFAC-6, PFAC-8) and variation of the process parameters time, input power and input mode (CW and PW). (3) Characterization of the aerogels surface and textural properties prior to and after the coating process.

## 2. Materials and Experimentation

### 2.1. Preparation of Aerogel Substrates

Aerogels based on alginate (Hydagen 500, BASF, Ludwigshafen, Germany), microcrystalline cellulose type II (JRS Pharma GmbH & Co. KG, Rosenberg, Germany, Vivapure^®^, 101), whey protein isolate (WPI, Agropur, BiPro 9500, Longueuil, Quebec, Canada) and potato protein isolate (PPI, Avebe, Veendam, The Netherlands, Solanic 200) were prepared via dissolution in water, gelation, solvent exchange with ethanol and supercritical drying with CO_2_, as detailed below. Different biopolymer contents were set for the different biopolymers (Table 1). An overall quantity of 500 g solution was produced from each biopolymer solution.

Dissolution of alginate, WPI and PPI was carried out by adding the educts to demineralized water followed by stirring at room temperature (500 rpm, overhead stirrer, 30 min). Dissolution of cellulose was carried out in aqueous NaOH solution at low temperature, following the procedure described in previous works [27]. The gelation of cellulose, WPI and PPI was induced thermally at different temperatures (Table 1) and gelation of alginate was carried out Ca^2+^-induced via internal setting method. Prior to gelation, biopolymer solutions were divided and transferred (5 mL each) into round cups (SecurTainer™ III, Thermo Fisher Scientific, Waltham, MA, USA) with a diameter of 28 mm, where the gelation took place. For thermal gelation, cups filled with solution were sealed and placed in a preheated oven for 30 min. Cellulose solutions showed color change from white to yellow during the gelation process. Obtained cellulose hydrogels were therefore washed with demineralized water until complete white gels were obtained. Ionotropic gelation of alginate via internal setting method was carried out as follows: The alginate solution was mixed with 0.25 wt% CaCO_3_ (g CaCO_3_/g alginate solution) via overhead stirrer (500 rpm, 30 min). In the next step, 0.4 wt% glucono-δ-lacton (GdL, Merck KGaA, g GdL/g alginate solution) was added to the alginate—CaCO_3_ solution, mixed for 2 min and quickly transferred into cups, before gelation occurred, in several minutes at room temperature. The so-formed weak gels were transferred to an aqueous CaCl_2_ solution (10 g CaCl_2_·2 H_2_O, Carl Roth GmbH & Co. KG, in 1 L demineralized water) in order to complete the gelation. For solvent exchange, alginate hydrogels were immersed stepwise in mixtures of ethanol (EtOH, Carl Roth GmbH & Co. KG, Karlsruhe, Germany) and water of different concentrations (30, 50, 70 wt% EtOH/water and 99.9 wt% anhydrous EtOH) until a minimum final EtOH concentration of 97.0 wt% inside the substrates was achieved (determined by density measurements, Anton Paar, DMA 4500 M).

Solvent exchange of WPI, PPI and cellulose hydrogels was carried out via direct solvent exchange by immersing the substrates in 99.9 wt% anhydrous EtOH until a minimum final concentration of 97.0 wt% ethanol inside the hydrogel slabs was achieved. After the solvent exchange, alcogels were placed in a high pressure autoclave with an overall volume of 3.9 L for supercritical drying with CO_2_. The supercritical drying was carried out at a temperature of 60 °C, pressure of 120 bar, under continuous flow of CO_2_ (flow rate = 120–160 g/min) through the autoclave until complete extraction of EtOH was achieved after 6 h. The dry aerogels were collected after slow depressurization (1 bar/min) of the autoclave and stored in sealed vessels in a desiccator over calcined silica gel prior to plasma coating.

### 2.2. Coating of Aerogel Particles via Cold Plasma Process

Cold plasma coating of aerogel substrates was carried out using a Tetra 100 plasma equipment provided by Diener Electronic GmbH & Co.KG, Ebhausen, Germany, with a maximum specific power of 300 W. The substrates were placed on a perforated metal plate that was subsequently fixed in the process chamber (dimensions: length and width = 400 mm, height = 625 mm) at a height of 100 mm. The temperature in the process chamber was adjusted to 40 °C, and a vacuum was set. Subsequently, the system pressure was adjusted to 0.15 mbar via introduction of argon into the chamber. Three different monomers were used: 1H,1H,2H,2H-Perfluordecyl-acrylat (PFAC-8), 1H,1H,2H,2H-perfluorooctylacrylate (PFAC-6) and Octafluorocyclobutane (C_4_F_8_). The liquid monomers PFAC-6 and PFAC-8 were injected into the chamber by a dosing pump with a constant flow rate of 30 µL/min. For the gaseous C_4_F_8_, a mass flow of 6 sccm was set. Each process was initiated by starting the glow discharge at the accorded input power and mode (CW or PW). Three different process settings were applied for polymerization of all monomers (Table 2). In PW mode, *t*_on_ and *t*_off_ times were set to 15 µs each, resulting in a duty cycle of 50%.

Different deposition times were set for all experiments in the range of 5–50 min by removing samples at the accorded process times. For this purpose, the process was briefly stopped and restarted after sampling.

### 2.3. Specific Surface Area, Mesopore Volume, Mesopore Diameter

Characterization of aerogels microstructural properties was carried out by low-temperature N_2_ adsorption-desorption analysis (Nova 3000e Surface Area Analyzer, Quantachrome Instruments, Boynton Beach, FL, USA). Parts of the monoliths were cut or broken into pieces, which included parts of the outer layer as well as the inner pore structure. An overall sample mass of 20–30 mg was used and all samples were degassed under vacuum at 60 °C for at least 6 h prior to each analysis. The Brunauer-Emmett-Teller (BET) method was used to determine the specific surface area *S_V_* and BET constant *C* as a single determination (plasma-treated samples) and double determination (non-treated samples). The relative standard of 5.8% (*S_V_*) and 15.1% (*C*) for the individual BET results were estimated from the average error of a 4-fold determination. The pore volume of the mesopores *V*_meso_ and mean pore diameter of mesopores *d*_meso_ were determined by the Barrett-Joyner-Halendia (BJH) method as single determinations under estimation of a relative measurment error of 5%.

### 2.4. Density and Porosity of Aerogels

The envelop density $\rho_{e}$ was determined by weighting aerogels monoliths on a fine balance (3-fold determination) according to:(2)$$\rho_{e}=\frac{m}{V}$$
with *m* being the sample weight and *V* the sample volume. Skeletal density *ρ_s_* was determined via helium pycnometry (Multivolume Micromeritics 1305, 4-fold determination) at room temperature. The overall porosity *ε* was calculated from envelope and skeletal densities as follows:(3)$$\varepsilon =\left(1-\frac{\rho_{e}}{\rho_{s}}\right)\cdot 100$$

### 2.5. Contact Angle Measurements

The static contact angle (STA) was determined using a drop shape analyzer (OCA 15EC) with the software SCA20 for OCA. All contact angles were taken with 1 µL (water) and 2 µL (*n*-hexadecane, Merck KGaA) of solution immediately after deposition of the droplet on the surface (detachment from syringe tip) and as average of contact angles at both sides. The standard error of 2.3% for the individual results was estimated from the average error of four double determinations. For the identification of sinking-in and deformation events, the drop was left on the surface for 60 s, and the respective effects were determined after this time.

### 2.6. Scanning Electron Microscopy and Energy Dispersive X-ray Spectroscopy

The surface properties and inner structure of the aerogels were characterized via scanning electron microscopy (SEM) and energy dispersive X-ray spectroscopy (EDS) (Zeiss Supra VP55, Jena, Germany). Samples were sputtered with a thin layer of gold (ca. 6 nm, Sputter Coater SCD 050, BAL-TEC) before analysis was started. The measurements were carried out under high vacuum at an accelerating voltage of 4–5 kV.

## 3. Results and Discussion

### 3.1. Properties of Non-Treated Aerogels

Aerogels were produced from four different biopolymers, whereas different porosities *ε* and envelope densities were obtained by variation of the solid content in the stock solutions (*c*_biop_). Specific surface areas were in the range of approximately 300 m^2^/g for polysaccharide gels and 200 m^2^/g for the protein-based materials (Table 3).

Increasing the solids content from 2.5 to 20 wt% resulted as expected in a linear increase (*R*^2^ 0.939) in the envelope density of the resulting aerogels, from 0.10 to 0.50 g/cm^3^, respectively. In contrast, the skeletal density decreased linearly with increasing solids content in the range of 2.5–15 wt% (*R*^2^ 0.977). The total porosity *ε* increased linearly (*R*^2^ 0.907) with decreasing *c*_biop_ (Appendix A). The contribution of micro-, meso- and macropores to the overall pore volume was determined as follows. The t-plot analysis of nitrogen desorption data showed that none of the aerogels obtained a significant amount of micropores. The macropore volume *V*_macro_ could therefore be calculated from *ρ_e_* and *V*_meso_:(4)$$V_{macro}=\frac{1}{\rho_{e}}-V_{meso}$$

The division of the total porosity into total macro- and mesopore volumes showed that both *V*_meso_ and *V*_macro_ were higher at lower *c*_biop_, while the ratio between the pore volumes *V*_macro_:*V*_meso_ changed as an exponential function of *c*_biop_: aerogels with lower *c*_biop_ contained higher macropore volume up to 83% at *c*_biop_ = 2.5 wt% (Figure 1f). These findings were also supported by SEM analysis, which provided qualitative insights into the macroporous aerogel structures: a continuous fibrous macroporous and interconnected pore network could be seen in alginate aerogels, whereas average macropores size range (237 ± 118 nm) could roughly be estimated based on image analysis (Figure 1a and Appendix A). In contrast, cellulose gels obtained a denser pore structure, which was interrupted by gaps in the size range of some micrometers (Figure 1b). Protein aerogels showed a generally denser network due to the higher biopolymer content: PPI gels consisted of a fibrous network with visible macropores in the size range of approximately 100 nm (Figure 1c), WPI gels formed a globular structures (Figure 1d). In the mesoporous range, unimodal pore size distributions were obtained, with significantly narrowest distribution and smallest diameter for the WPI aerogel (Figure 1e).

Gels derived from different biopolymers exhibited different surface polarities: This was reflected in water contact angle measurements of the pure aerogels, which led to immediate collapse of the gel when being exposed to a liquid water droplet in the case of the highly polar polysaccharide gels. Gelation of both WPI and PPI was influenced by hydrophobic interactions between hydrophobic amino acids side chains [5,28], which contributed to an overall lower surface polarity. Therefore, static water contact angles could be determined for non-treated protein gels (83 ± 2°), which showed immediate discoloration after application of the droplet, while no collapse and significant deformation occurred. Insights about the surface polarity in the pores can be provided via the BET *C*-constant, which was exponentially related to the energy of monolayer adsorption. High values of *C* (~150) are generally associated with adsorption on high-energy surface sites (in the absence of micropores), whereas the overall value of the *c*-constant decreases in presence of increasing non-polar moieties, as shown for silica-casein hybrid aerogels by Lazar et al. [29,30] In this work, the order of *C*-constant values (alginate = 148 > cellulose = 97 > WPI and PPI = 46) corresponded well with the expected surface energy of the aerogels.

### 3.2. Liquid Repellency of Coated Aerogels

A systematic screening was conducted to investigate the effects of cold plasma process time *t* (*t* = 5–50 min), input mode and input power in three different process modes (PW, *p*_input_ = 300 W; CW, *p*_input_ = 90 W; CW, *p*_input_ = 30 W) on the surface wettability of all different combinations of aerogel substrates and coating materials (C_4_F_8_, PFAC-6, PFAC-8). Surface water wettability was tested via determination of the static contact angle θ of water for all samples. Four different cases could be distinguished after deposition of water droplets on the aerogel substrates: (1) Immediate sinking in of the droplet and collapse of the gel at the site of application in the case of polysaccharide gels (unsuccessful process setting). (2) Immediate discoloration of the gels surface in the case of protein gels (unsuccessful process setting). (3) Deformation of the substrate at the site of application during the measurement, resulting in formation of a dimple and quick change of the obtained contact angle over time, whereas no absorption of the droplet or destruction of the aerogel took place. (4) No deformation at the droplet application site and stable contact angle over a timeframe of ≥1 min. Two process times were defined accordingly: *t*(1) as the minimum process time, after which a contact angle could be measured without substrate wetting or destruction, but under substrate deformation (case 3), and *t*(2) as the minimum process time, which was needed to obtain contact angles without further deformation of the substrate over time (case 4). Water surface wettability was achieved for all combinations of substrates and coatings materials and resulted in values of θ in a broad range from 78–154°, whereas deformation of the substrate occurred only in the case of the polysaccharide aerogels (Figure 2, Table 4).

The latter can be explained by the different mechanical stabilities of the gels: Protein gels with comparably high solids content showed a rigid network and no plasticity. In contrast, alginate gels were most susceptible against deformation or pore collapse since they exhibited the highest intrinsic surface energy and lowest solids content. We conclude that a sufficient thickness of the polymer layer is needed in case of polysaccharide aerogels to reach the time *t*(2). Significant surface hydrophobization of all combinations of substrate coating material was achieved after a short processing time *t*(1) = 5 min. Coating with C_4_F_8_ in CW mode and *p*_input_ = 90 W resulted in the highest contact angles in the case of cellulose, PPI and WPI (Table 4). In addition, process times of *t*(2) = 10 min were sufficient in preventing alginate and cellulose aerogels deformation in the same mode.

Stable layer formation was also achieved in PW/*p*_input_ = 300 W mode after a significantly higher process time of *t*(2) = 50 min. CW plasma polymerization generally led to higher monomer fragmentation and crosslinking degree of the resulting polymer compared with PW mode and was reported to favor the formation of low energy CF_3_ groups in the case of C_4_F_8_ polymerization, while PW plasma film thickness rate was significantly lower [9,13]. Our results show accordingly faster formation of stable layers on polysaccharide aerogels in CW/*p*_input_ = 90 W mode of C_4_F_8_ polymerization. In contrast, deformation of polysaccharide aerogels was noticed at *p*_input_ = 30 W at all process times. The observed trend may be attributed to changes in monomer deposition rate since deposition rate was found to increase with rising average power under CW conditions [31]. In contrast, higher input power of 90 W did not lead to the formation of stable layers in the case of CW PFAC-8 polymerization and led to delayed *t*(2). A reversed trend depending on deposition rate on input power was also reported by Coulson et al. for PW cold plasma polymerization of 1H,1H,2H,2H-heptadecafluorodecyl acrylate and was consistent with fragmentation and damage of the long perfluoro-alkyl chains [32]. Lower influence of process parameters was observed for PFAC-6 monomer with shorter perfluoro-alkyl chains, whereas CW/*p*_input_ = 90 W mode provided similar results compared with the other modes and resulted in stable layer formation on cellulose aerogels with *t*(2) = 20 min. Summarized, preferable process conditions to achieve liquid water repellency were determined for each coating material. The conditions were found to depend on the process parameters as well as on the nature of monomers and substrates.

Oleophobic repellency was tested by determination of the static contact angle θ_ol_ using *n*-hexadecane as probe liquid. Protein aerogels are well known for their good oil uptake capability [33]. Immediate absorption of n-hexadecane droplets was therefore observed in case of non-treated gels and also of poly-C_4_F_8_ coated gels, since longer perfluorinated chain lengths are necessary to obtain oleophobic behavior [32]. PFAC-6 (PW, *p*_input_ = 300 W) and PFAC-8 (CW, *p*_input_ = 30 W) coatings resulted in repellency of *n*-hexadecane with a range of θ_ol_ = 71–134° on all tested substrates after short process times of 5 min (Table 4). In accordance with the results for water repellency, no general influence of the process time was determined, while the value of θ_ol_ depended mainly on the substrate/coating combination (Figure 3). Significantly higher values of θ_ol_ were obtained by application of poly-PFAC-6 on protein aerogels compared with poly-PFAC-8 on protein aerogels, while no differences were found in case of polysaccharide aerogels (Figure 3).

It is notable that no deformation of polysaccharide substrates took place after application of n-hexadecane droplets. Therefore, substrate deformation did not result from the droplets weight: influences such as the electrostatic attraction between droplet and substrate (polar–polar interactions) may play an important role in the deformation process and require a separate investigation.

### 3.3. Coating Properties

The surface microstructure and homogeneity of the coating layers were qualitatively analyzed via SEM. Film quality and structure depended on the individual combinations of process parameters, substrate/coating materials combination as well as the process time as exemplified in Figure 4. Additional examples are provided in the Appendix A.

Application of C_4_F_8_ in CW/*p*_input_ = 90 W mode on alginate aerogels resulted in the fast formation of homogeneous surface coating layers after short process times of 5 min, while cellulose and PPI aerogels were not completely covered (cf. Figure 4a,c,d). Increasing the process time to 50 min resulted in generally smooth coatings with visible aggregates on the top for alginate and PPI aerogels, while inhomogeneities were observed on the coated cellulose aerogel (Figure 4a,c,d). WPI aerogels exhibited non-covered regions with open pores, even after the maximum process time of 50 min (Figure 4d), which speaks to a generally insufficient adhesion between coating materials and WPI. The results indicate that initially homogeneous aerogel surfaces without micro-gaps in the pore structure and high polarity of the substrates were beneficial for the fast formation of uniform coating layers. Furthermore, we can conclude that complete film formation is not necessarily required to obtain water repellent surfaces, since contact angles in the range of approximately 120–140° were also determined (under substrate deformation) in cases of non-complete coverage of the pores (see examples Appendix A). In contrast with C_4_F_8_-based coatings, poly-PFAC-6 and PFAC-8 showed rougher surfaces with crystallization of the material in some cases (Figure 4e–g), whereas a generally higher degree of crystallization was found for poly-PFAC-8. This observation may be explained by the different side chain lengths of the PFAC-monomers, which resulted in different reorganization and crystallization properties, leading to enhanced crystallization and formation of ordered structures for side chain lengths ≥8 [34].

### 3.4. Textural Properties

Quantification of the influence of the cold plasma coating process on the microstructure was possible by comparing the specific surface area *S_V_* of aerogels before and after post-modification: The decrease in the specific surface area could principally result from pore collapse or from the contribution of additional mass from non-porous coating material. Both effects were expected to be non-significant in the case of cold plasma polymerization due to the small amount of added material and absence of liquids and moisture during the process.

A negative effect of the coating process on the specific surface area was indeed not detected, showing that the porous structure remained intact during the process under all process conditions. In contrast, a significant increase in Δ*S_V_* in specific surface areas of treated substrates up to Δ*S_V_* = +179 m^2^/g (corresponding to an increase up to +61%) was determined (Figure 5). The average change of *S_V_* depended strongly on the aerogel nature (+ ΔS_v_ alginate > cellulose > PPI > WPI), whereas the individual combinations with coating materials, the process settings and the process time also played important roles. A generally equal influence of the process time was found for all individual settings: The maximum Δ*S_V_* was reached after a process time of 10 min, followed by steady state conditions in the range of *t* = 10–40 min and a slight decrease when running the process for 50 min. Furthermore, the highest changes occurred using C_4_F_8_ monomer in PW/*p*_input_ = 300 W mode in all cases, which is therefore presented separately (circles in Figure 5).

The detected increase in the BET surface area was a clear indication of polymerization taking place inside of the pores, resulting in the formation of new, hydrophobic zones that could be detected with nitrogen adsorption experiments. This was also reflected in a lowering of the BET *C* constant compared with the pristine polysaccharide aerogels, which is indicative of the deposition of non-polar coating material in the pores (Figure 5). The maximum change in specific surface area Δ*S_V_* was quantified for each individual setting by averaging the values of *S_V_* in the steady state range from *t* = 10–40 min. The Δ*S_V_* was related to the porous structures of the different substrates and increased exponentially (*R*^2^ 0.902–0.999) with the overall porosity and linearly (*R*^2^ 0.984–0.999) with the overall pore volume of the aerogels (Appendix A). An exponential increase in Δ*S_V_* by increasing the mesopore volume and a linear increase by increasing the macropore volume show the significance as well as individual contributions of differently sized pores in the process (Figure 6). We surmise that an interconnected macroporous network, such as that observed for alginate aerogels (see Figure 1a), should contribute to a relatively unhindered transport of activated monomer into inner aerogels parts but should also provide comparably low surface area for the coating processes (transport-pores). In contrast, mesopores provided high specific surface area on which reactions with activated monomer could occur, but could only transport the activated monomer into the aerogel to a limited extent (reaction-pores).

Transport of coating material into the pores was verified via EDS spectroscopy of cut-open substrates, an example of which was carried out for C_4_F_8_-coating. The thin coating layer and the inner porous part were clearly distinguishable on the corresponding SEM images and the distance *d* (max. 112 µm) from the outer layer was determined accordingly (Figure 7 left). The amount of fluorine was calculated as relative abundance *r*_F_ in relation to oxygen and carbon. Significant amounts of fluorine were detected in the inner part of the aerogels, which is a proof of the penetration of monomer molecules into the substrates. The overall fluorine content in the pores was related to the distance *d* from the outer layer, while the initial value of *r*_F_ on the surface varied significantly with the aerogel nature (Figure 7, right). The latter can be expected to depend mainly on the adhesion between monomer and substrate, resulting in significantly higher values of *r*_F_ for the more polar polysaccharide aerogels. The exponential decay of *r*_F_ spoke to a significant mass transport limitation of monomer diffusion into the bulk of the aerogels. The value of *r*_F_ remained mostly constant in the inner part of the substrates.

The alginate aerogel shows a higher fluorine content in the inner part (*r*_F_ = 11.2 ± 1.6 wt% at *d* ≥ 50 µm) and a lower slope of *r*_F_-decay as compared to the cellulose aerogel. Our results suggest that a high overall porosity and macroporous content are in case of good monomer adhesion decisive for the monomer transport into the material. In case of a lower adhesion of the monomer (the case of protein aerogels), transport in the pores is relatively unhindered on the one hand, but the overall fluorine content on the aerogels surface and in the pores is significantly lower when compared to the alginate aerogel on the other hand. Summarized we conclude that an interconnected macroporous network combined with high surface adhesivity are both necessary to achieve a good monomer transport into the materials on the one hand, while enabling sufficient monomer deposition onto the pore walls on the other hand. 

Changes of the porous structure caused by the plasma deposition of the coating materials were determined via BJH desorption analysis. Pore size distributions of alginate aerogels showed that smaller pores with a range of pore radius *r*_pore_ ≈ 3–7 nm disappeared after a short process time of 5 min, while new mesopores with *r*_pore_ ≈ 15–25 nm were generated (Figure 8 top, Appendix A). The generation of new pores in the same size range was observed for cellulose aerogels, whereas the distributions generally broadened, and additional pore volume was also obtained in the range of low *r*_pore_ (Figure 8 bottom, Appendix A).

It is of interest to try elucidating the mechanism of the coating given the diversity in nature and properties of the studied aerogels. Although a complete elucidation of all steps involved in the cold plasma coating of aerogels is beyond the scope of this study, we discuss here probable elementary processes that stack up the entire coating process and exclude unlikely ones. We begin by building upon the observed increase in the *specific* surface area. Because the nitrogen porosimetry can detect pores with *r*_pore_ ≲ 75 nm, one may speculate that the increase in the specific surface area in the range of a processing time of 10–40 min is due to the deposition onto the macropores (*r*_pore_ > 75 nm), which now become detectable by N_2_ porosimetry. To test out this hypothesis, we estimated the amount of the material required to convert all macropores into mesopores as follows. We consider here an extreme case where all macropores are converted to mesopores. The mesopores remain however non-penetrable to the monomer, i.e., their contribution to the total specific surface area does not change during the plasma coating.

When coating material with a mass m is deposited in macropores, the macropore volume of the coated aerogel is
(5)$$V_{macro}=V_{macro,0}-m/\rho_{sk}$$
where $V_{macro,0}$ is the macropore volume of the pristine aerogel, and $\rho_{sk}$ is the skeletal density of the coating (assumed to be equal 2.2 g/cm^3^ as for Teflon). In the following, the subscript “0” stands for pristine aerogel. For our purposes it is sufficient to treat a macroporous aerogel network within the cylindrical pore model [35]. According to this model, there is one efficient cylindrical pore with a diameter $d_{pore}$ and length $h_{pore}$. The macropore volume is then given by
(6)$$V_{macro}=\pi /4d_{pore}^{2}h_{pore}$$

The specific surface area in the cylindrical pore model can be easily calculated as [27]
(7)$$S_{V}=4V_{macro}/d_{pore}$$

For example, pristine alginates have $V_{macro}$ = 8.6 cm^3^/g (see Figure 6) and the average macropore size is $d_{pore}$ = 237 nm (see Section 3.1), which yields via Equation (7) a value $S_{V,0}$ = 145 m^2^/g. This value corresponds to non-detectable specific surface area due to macropores only. If we then assume that the pore length $h_{pore}$ does not change significantly during the plasma coating, from Equation (6) we readily have
(8)$$d_{pore}=d_{pore,0}\sqrt{V_{macro}/V_{macro,0}}$$

Substituting Equations (5) and (8) in Equation (7) we finally obtain for the specific surface area of the coated aerogel:(9)$$S_{V}=\frac{S_{V,0}}{m_{0}}\frac{1}{m+m_{0}}\sqrt{1-\frac{m}{\rho_{sk}V_{macro,0}}}$$

Because the *specific* surface area (m^2^/g) is what is always measured, $m_{0}$ = 1 g in Equation (9). Analysis of Equation (9) shows that the specific surface area $S_{V}$ is a decreasing function of the deposited mass m. Therefore, the specific surface area can only decrease in the coating process, i.e., $\Delta S_{V}=S_{V}-S_{V,0}$ is always negative. Results for changes of a specific surface contradict this picture since $\Delta S_{V}$ is positive (see Figure 5 and Figure 6). Therefore, we are forced to search for alternative mechanisms.

One probable explanation for the observed increase in the specific surface area is that the polymer deposition takes place where the mass transport through the porous network becomes limited, e.g., where two pores with significantly different diameters $d_{1}$ and $d_{2}$ are coupled to each other (Figure 9). Deposition of a film with a thickness h would create and extra surface $\pi d^{2}/2$ at each coupling point (given that $d_{1}<d_{2}$). The total surface area for N coupling points is
(10)$$S_{V,0}m_{0}+N\pi d^{2}/2$$
whereas the total increase in mass is
(11)$$m_{0}+\rho_{sk}N\pi d^{2}h/4$$

The deposited polymer layer can in principle be mesoporous with its own specific surface area of $S_{V,coat}$. This is what would add an extra term $S_{V,coat}\rho_{sk}N\pi d^{2}h/4$ in Equation (10).

The ratio of the expressions Equations (10) and (11) gives the specific surface area of the coated aerogel. Calculations show that for thin layers (~1 nm), the gain in surface area overweighs the gain in the coating mass. Therefore, this mechanism agrees qualitatively with our experimental observations. Further insights are provided via SEM images of the pore structures from pristine and coated aerogels, wherein coating material and alginate fibers are clearly distinguishable (Figure 10a,b). High resolution pictures indicate that the deposited polymer layer has an intrinsic porosity: New adsorption sites are provided by narrow crevices and mesopores formed by growing coating material (Figure 10c,d and Appendix A). Furthermore, change of alginate fiber thickness due to deposited coating material could principally cause additional changes of the specific surface area in case of mesopores, as schematically indicated in Figure 9. Since any deposition of coating material layers onto fibrils leads also to a drop of the specific surface area (no matter how thin the layer is), we come to the conclusion that the only ways to create additional surface are as mentioned above: the deposition of thin films in pore coupling points and by intrinsic porosity of the polymer layers.

Quantitative comparisons going beyond the basic calculations presented above are not possible at the moment since the parameters N, h and $S_{V,coat}$ are unknown. Elucidation of a quantitative model of the cold plasma coating is a part of our ongoing work. Generally, we believe that cold plasma coating is a very well suited method for the post-processing of an aerogel surface and worth further investigations—for instance, by extension to additional aerogel types and monomers.

## 4. Conclusions

It was shown first that cold plasma coating with fluorinated monomers is a suitable method for fast, simple and material-efficient hydrophobic and oleophobic post-modification of various open porous biopolymer aerogels with high specific surface areas (200–300 m^2^/g). Significant effects were achieved after short process times of 5–10 min, and no further purification of the substrates was necessary. The wide applicability of the process was demonstrated by coating four different aerogels with different polarities and total porosities (64–95%) as well as macro-to-mesoporous fractions. It was possible to coat all substrates with layers of three different fluorinated coatings (C_4_F_8_, PFAC-6, PFAC-8), resulting in hydrophobic to superhydrophobic liquid water-repellent surfaces, which showed static water contact angles up to 154°. While application of water droplets on non-treated alginate and cellulose gels resulted in immediate pore collapse at the site of droplet application, protection of the gels was generally achieved after process times of 5 min. The individual results depended on the specific monomer/substrate pair and the process parameters (input power, input mode and process time). The polymerization of perfluorinated acrylates resulted in oil repellent surfaces, as verified by contact angle measurements with n-hexadecane. In many cases, coated polysaccharide aerogels showed a slight deformation at the site where water droplet was applied. Prevention of this effect was achieved via C_4_F_8_ polymerization in CW mode and input power of 90 W after process times ≥10 min.

Furthermore, it was demonstrated that new adsorption sites in the pores were formed during the process, resulting in a significant increase of up to 61% in the specific surface area. Maximization of Δ*S_V_* was achieved in the PW mode, using C_4_F_8_ as monomer. A steady state was reached after *t* = 10–40 min, resulting in constant values of Δ*S_V_* under the given process conditions. Relations of the Δ*S_V_* with the overall aerogel porosity as well as individual macro- and mesopore volumes were identified. It was shown that high porosities ≥ 90% are necessary for achieving high effects of Δ*S_V_* ≥ 100 m^2^/g. While changes of the BET C constant indicated the generation of moieties with lower surface energies in polysaccharide gels, the penetration of coating material into the pores was proven via EDS analysis of cut-open substrates. The results indicate that two main effects determine the intrusion depth and final fluorine concentration in the pores: (1) the adhesivity of the substrate material to the activated monomers and monomer fragments; (2) the composition of the porous structure, whereas an interconnected, highly macroporous network is beneficial for a good transport of monomer into the structure. The dynamic changes of the mesoporous pore structure during the process, as determined via BJH desorption analysis, led in the case of alginate aerogels to a decrease in pore volume from small mesopores in the range of *r*_pore_ ≤ 10 nm and an increase in the range of *r*_pore_ approximately 15–60 nm. Estimations based on geometric and mass balance aspects showed that the observed increase in the specific surface area may result from two processes: (i) polymer deposition at coupling points of macro- and mesopores and (ii) due to intrinsic porosity of the deposited polymer layer.

Summarized, it was shown that surface plasma coating of biopolymer aerogels with fluorinated monomers leads to modification of the external surfaces as well as to modification of the pore structure of aerogels. An increase in the specific surface area via coating processes has not yet been reported and may be a significant step towards further improvement of biopolymer aerogel properties. While the results of this study provide first considerations about the key process parameters and material properties for pore modification, additional systematic studies are necessary in order to quantify the influences of power input and input mode from the process side and substrate adhesivity and pore structure from the materials side.

## Figures and Tables

**Figure 1 polymers-13-03000-f001:**
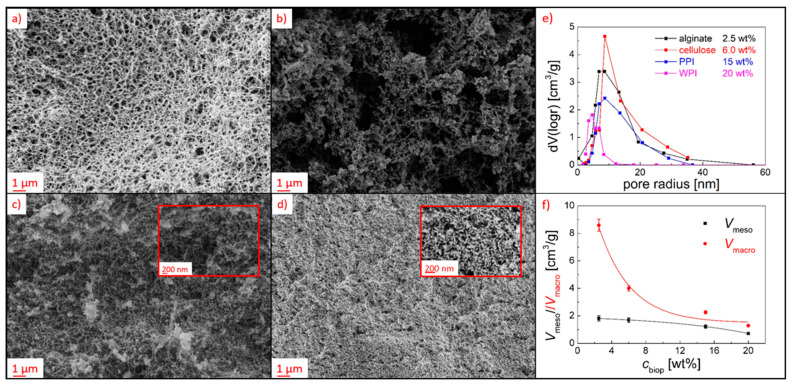
(**a**–**d**) SEM pictures of non-treated aerogels surfaces: (**a**) alginate, (**b**) cellulose, (**c**) PPI, (**d**) WPI, (**e**) pore size distributions, (**f**) macro- and mesopore volumes.

**Figure 2 polymers-13-03000-f002:**
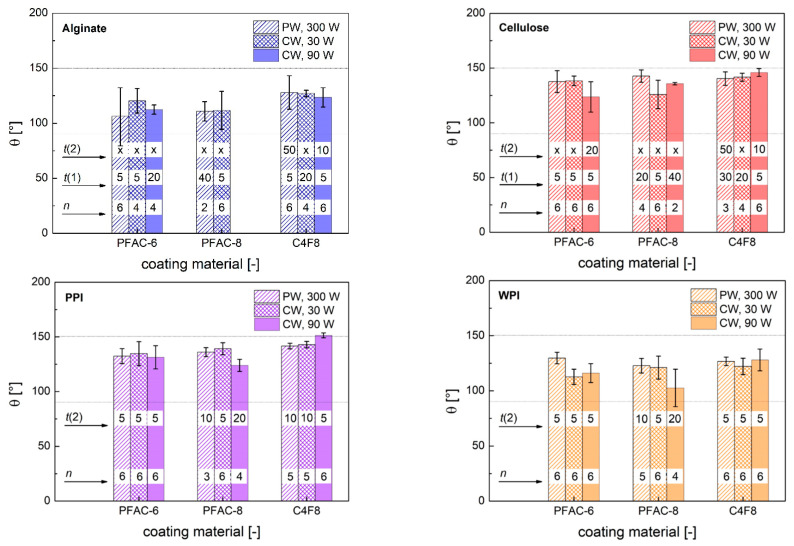
Static water contact angles θ depending on coating material and aerogel type. Dashed lines mark the regions of hydrophobicity (θ = 90–150°) to superhydrophobicity (θ ≥ 150°). All values represent the average of successful settings *n* (cases 3 and 4) at different process times; error bars represent the standard deviation of the averaged values. PW (pulsed wave mode) and CW (continuous wave mode) stand for the power input mode; accordingly, numbers represent the input energy. Numbers in columns stand for the minimum process times (min) needed to achieve the corresponding case; x denotes that the corresponding case did not occur under the given process conditions.

**Figure 3 polymers-13-03000-f003:**
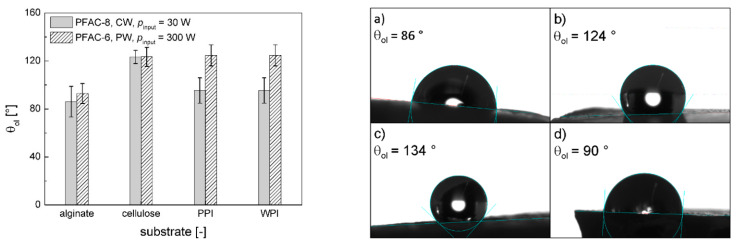
θ_ol_ for different substrate/coating combinations. Values in the overview (**left**) represent the average at different process times under a given process condition (as indicated in the legend); error bars represent the standard deviation (*n* = 6). Raw data (**right**) represent exemplary process settings: (**a**) alginate, t = 5 min, PFAC-8, CW, p_input_ = 30 W, (**b**) cellulose, *t* = 10 min, PFAC-6, PW, *p*_input_ = 300 W, (**c**) PPI, *t* = 5 min, PFAC-6, PW, *p*_input_ = 300 W, (**d**) PPI, *t* = 5 min, PFAC-8, CW, *p*_input_ = 30 W.

**Figure 4 polymers-13-03000-f004:**
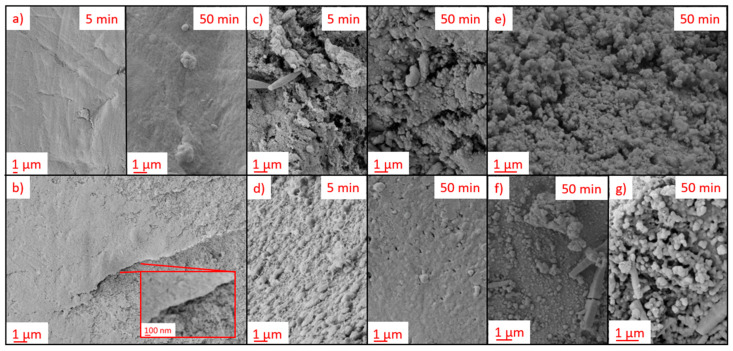
SEM pictures of external coating on different aerogels: (**a**) alginate, C_4_F_8_, CW, *p*_input_ = 90 W, (**b**) WPI, C_4_F_8_, CW, *p*_input_ = 90 W, (**c**) cellulose, C_4_F_8_, CW, *p*_input_ = 90 W, (**d**) PPI, C_4_F_8_, CW, *p*_input_ = 90 W, (**e**) alginate PFAC8, CW, *p*_input_ = 30 W, (**f**) PPI, PFAC-6, CW, *p*_input_ = 90 W, (**g**) PPI, PFAC-8, CW, *p*_input_ = 30 W.

**Figure 5 polymers-13-03000-f005:**
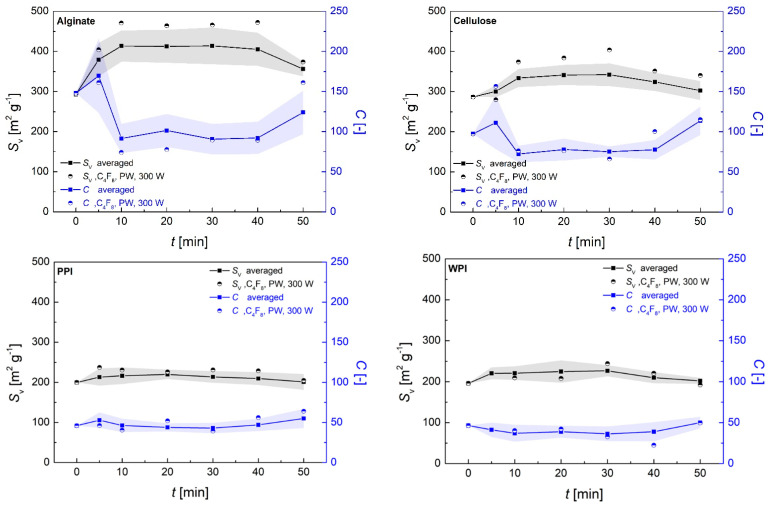
Changes of specific surface area and C-constant depending on process time and process conditions. Circles represent the values obtained with C_4_F_8_ monomer in PW mode with *p*_input_ = 90 W. Squares represent the averaged values from all coating materials and process conditions, excluding PFAC-8 coating in CW mode with *p*_input_ = 300 W. Error areas represent the standard deviation from averaged values; solid lines are drawn to guide the eyes.

**Figure 6 polymers-13-03000-f006:**
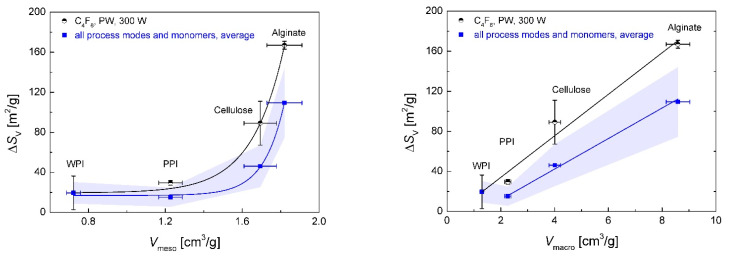
Increase in Δ*S_V_* depending on the mesopore volume (**left**) and the macropore volume (**right**). Circles represent the values obtained with C_4_F_8_ monomer in PW mode with *p*_input_ = 300 W. Squares represent the averaged values from all coating materials and process conditions, excluding PFAC-8 coating in CW mode with *p*_input_ = 90 W. Straight lines represent exponential (**left**) and linear (**right**) fitting. Error areas represent the standard deviation of Δ*S_V_* from averaged values at different process conditions; error bars represent the relative measurement errors (*x* error) and standard deviation of Δ*S_V_* for the individual process condition (*y* error).

**Figure 7 polymers-13-03000-f007:**
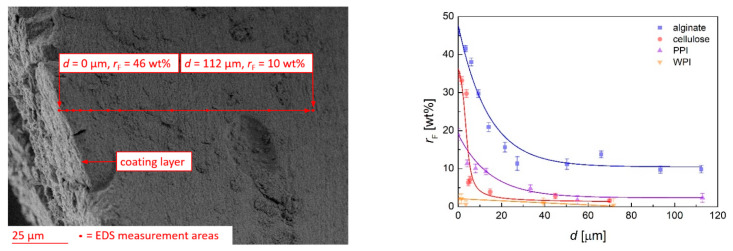
SEM picture of cut-open alginate aerogel substrate with external coating and inner porous part (**left**). Points mark the areas according to EDS measurements. Fluorine content depending on the distance from outer layer (**right**). Error bars represent the error of the individual measurements; solid lines illustrate the course of *r*_F_ for different aerogels. All measurements were performed for aerogel substrates treated under the following process condition: C_4_F_8_, CW, *p*_input_ = 90 W, *t* = 50 min.

**Figure 8 polymers-13-03000-f008:**
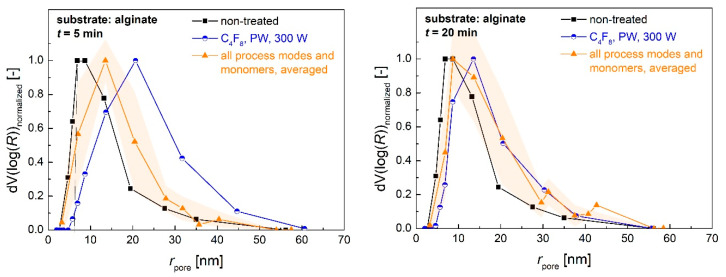
Comparison of BJH pore size distributions (normalized) from alginate (**top**) and cellulose (**bottom**) aerogels prior to and after the plasma coating process. Round data points represent the values obtained after coating with C_4_F_8_ in PW mode with *p*_input_ = 300 W; triangular data points represent the averaged values obtained after all other coating process conditions, excluding PFAC-8 coating in CW mode with *p*_input_ = 90 W. Error areas represent the standard deviation from averaged values.

**Figure 9 polymers-13-03000-f009:**
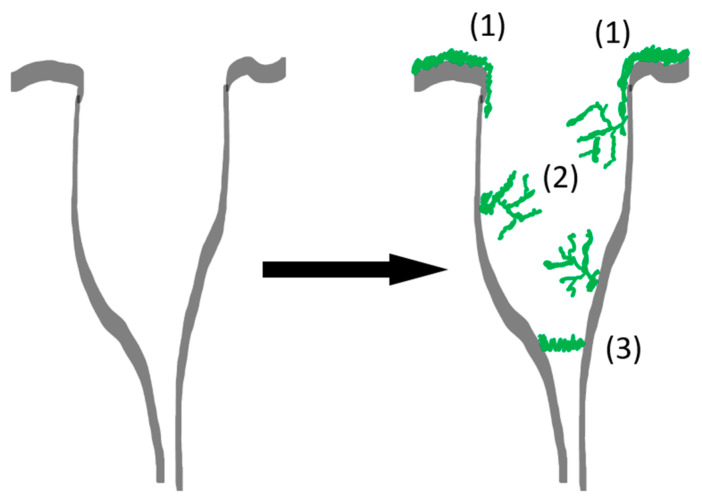
Schematic representation of two coupled pores. The large pore (macropore) is exposed to the gas phase with activated monomer (not shown). A layer at the outer surface (1), porous structures from the deposited polymer (2) and a thin layer at the coupling point of two pores (3) are shown. The layer (1) does not lead to an increase in the specific surface area, while the structures (2) and (3) do. Deposition of the layer (3) may be due to a limited mass transfer through the smaller pore.

**Figure 10 polymers-13-03000-f010:**
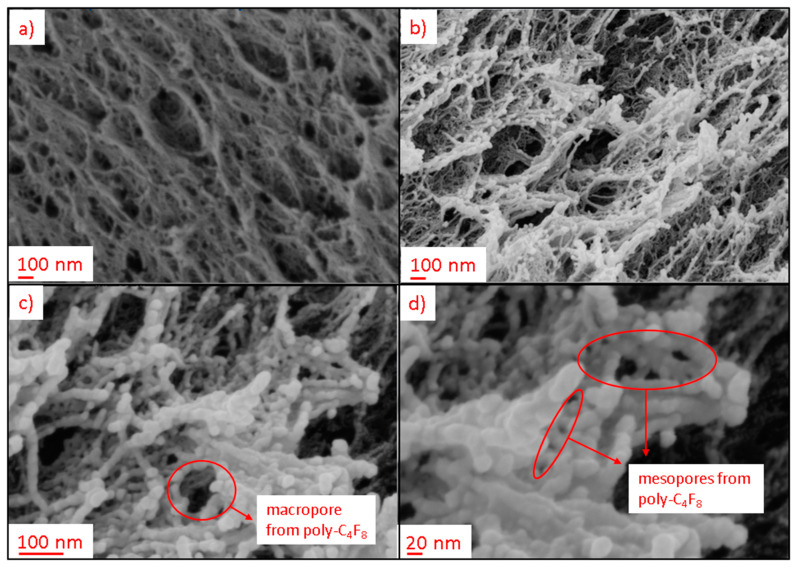
SEM pictures: (**a**) porous structure of non-treated alginate aerogel; (**b**) porous structure of treated alginate aerogel, coating material C_4_F_8_, CW, *p*_input_ = 90 W, *t* = 50 min, distance from outer layer = 134 µm; (**c**) and (**d**) high-resolution pictures of poly-C_4_F_8_ in alginate aerogel and new pores generated thereof.

**Table 1 polymers-13-03000-t001:** Biopolymer content and gelation conditions for production of hydrogels from different biopolymers.

Material [–]	Biopolymer Content (wt%)	Gelation Method [-]	Post Treatment After Hydrogel Formation [-]
alginate	2.5	ionotropic in presence of Ca^2+^	completion of gelation with aqueous CaCl_2_ solution
cellulose	6.0	Thermal 60 °C, 30 min	washing with demineralized water
PPI	15	Thermal 85 °C, 30 min	none
WPI	20	Thermal 90 °C, 30 min	none

**Table 2 polymers-13-03000-t002:** Process modes used for plasma polymerization of different monomers.

Mode [-]	*p*_input_ (W)	Duty Cycle (%)	Monomer [-]	Flow Rate [-]
PW	300	50	C_4_F_8_	6 sccm
CW	90	100	PFAC-6	30 µL/min
CW	30	100	PFAC-8	30 µL/min
PW	300	50	C_4_F_8_	6 sccm
CW	90	100	PFAC-6	30 µL/min
CW	30	100	PFAC-8	30 µL/min
PW	300	50	C_4_F_8_	6 sccm
CW	90	100	PFAC-6	30 µL/min
CW	30	100	PFAC-8	30 µL/min

**Table 3 polymers-13-03000-t003:** Properties of non-treated aerogels produced from different biopolymers.

Material [-]	*c*_biop_ (wt%)	*S_V_* (m^2^/g)	*d_meso_* (nm)	*V*_meso_ (cc/g)	*V*_macro_ (cc/g)	ρ_s_ (g/cm^3^)	ρ_e_ (g/cm^3^)	ε (%)	*C* [-]
Alginate	2.5	301 ± 18	13.7 ± 0.7	1.82 ± 0.09	1.898 ± 0.020	0.0961 ± 1·10^−4^	1.90 ± 2·10^−2^	95.0 ± 1.0	148 ± 22
Cellulose	6.0	289 ± 17	17.4 ± 0.9	1.69 ± 0.09	1.613 ± 0.049	0.1754 ± 6·10^−4^	1.61 ± 5·10^−2^	89.0 ± 3.0	97 ± 15
PPI	15.0	200 ± 12	13.9 ± 0.7	1.23 ± 0.06	1.345 ± 0.012	0.2865 ± 3·10^−4^	1.34 ± 1·10^−2^	78.7 ± 0.8	46 ± 7
WPI	20.0	201 ± 12	7.1 ± 0.4	0.72 ± 0.04	1.357 ± 0.009	0.4956 ± 2·10^−4^	1.36 ± 1·10^−2^	63.5 ± 0.5	46 ± 7

**Table 4 polymers-13-03000-t004:** Range of static water contact angles on different aerogel substrates.

Aerogel [-]	Range of θ (°)	Raw Data θ Maximum [-]	Conditions Maximum θ [-]	Conditions for Non-Deformed Substrates [-]
alginate	78–139	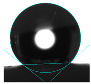	*t* = 10 min, C_4_F_8_PW, 300 W	C_4_F_8_, CW, 90 W, 10 minC_4_F_8_, PW, 300 W, 50 min
cellulose	102–151	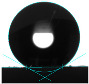	*t* = 40 min, C_4_F_8_CW, 300 W	C_4_F_8_, CW, 90 W, 10 minC_4_F_8_, PW, 300 W, 50 minPFAC-6, CW, 90 W, 20 min
PPI	116–154	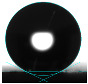	*t* = 5 min, C_4_F_8_CW, 90 W	all
WPI	103–142	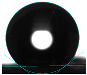	*t* = 10 min, C_4_F_8_CW, 90 W	all

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
