# Peer review of "Hydrophobic Modification of Biopolymer Aerogels by Cold Plasma Coating"

_polymers, 2021, doi:10.3390/polym13173000_

Round 1
Reviewer 1 Report
The paper entitled "Hydrophobic Modification of Biopolymer Aerogels by Cold Plasma Coating" by Baldur Schroeter and co. describes the deposition of protective hydrophobic and oleophobic polymer layers on hydrophilic biopolymer aerogels by cold plasma polymerization
The paper is well strutured, the preparation methods and the results are clearly presented.
The selected characterisation methods are appropriate for the selected sample and the discussions are clear and well organised.
The Conlusion part presents the novelty of the work
The only issue which can be improved, is to demonstrate the non-toxicity of the obtained materials as biopolymers, like is written in Introduction part: page 3, row 113 A promising non-toxic alternative to the use of chloromethanes and chlorosilanes is the application of fluorocarbon based polymer coatings, since they exhibit extremely low surface energies."
Author Response
We thank Reviewer 1) for the work and are happy about the overall positive feedback of our manuscript.
Below our answer to the comment “The only issue which can be improved, is to demonstrate the non-toxicity of the obtained materials as biopolymers, like is written in Introduction part: page 3, row 113 A promising non-toxic alternative to the use of chloromethanes and chlorosilanes is the application of fluorocarbon based polymer coatings, since they exhibit extremely low surface energies."
Answer to Reviewer:
Fluorcarbon based polymer coatings are in principle non-toxic, since they have high thermal, chemical, photochemical, hydrolytic, and biological stability. This accounts for Teflon-like layers derived from C4F8 as well as polyfluoroalkyl substances.
Since polyfluoroalkyl substances (PFAS) are a large, diverse group of substances that, in some respects, challenge easy distinction for assessment and management, a clearer understanding of the origin of PFAS found in the environment and assessment of their properties is needed to be able to determine which classes of PFAS require management action. Per- and polyfluoroalkyl substances must be assessed taking into account their differences in chemical, physical, thermal, and biological properties. A single, globally harmonized system for PFAS classification has not yet been defined, resulting in a lack of distinction between PFAS.
A recent review (A Critical Review of the Application of Polymer of Low Concern and Regulatory Criteria to Fluoropolymers, new literature in manuscript,[25]) brings together fluoropolymer toxicity data, human clinical data, and physical, chemical, thermal, and biological data for review and assessment and shows, that fluoropolymers satisfy widely accepted assessment criteria to be considered as “polymers of low concern” (PLC). Details about the distinct rules (based on region) and actual classifications (based on applications) are discussed.
In this regard, we added a short statement and reference to the manuscript (line 115-116, page 3):
Fluoropolymer layers satisfy the widely accepted regulatory assessment criteria to be considered as “polymers of low concern” .[25]
Reviewer 2 Report
This interesting manusript reports on the hydrophobic and oleophobic cold plasma modification of aerogels surface. This works analyzing wetting properties, morhology and homogeneity of the coating.
The only question to authors from my side is;
Any concerns regrading the stability and toxicity of such modifications? And what about the security of the process, especially if one would like to upscale the process? What are the toxicities of outgas and precursors?
Author Response
We thank Reviewer 2) for the work and are happy about the overall positive feedback of our manuscript.
Below our answers to the comment “The only question to authors from my side is; Any concerns regrading the stability and toxicity of such modifications? And what about the security of the process, especially if one would like to upscale the process? What are the toxicities of outgas and precursors?”
Answer regarding stability and toxicity of the modifications
Fluorcarbon based polymer coatings are in principle non-toxic, since they have high thermal, chemical, photochemical, hydrolytic, and biological stability. This accounts for Teflon-like layers derived from C4F8 as well as polyfluoroalkyl substances.
Since polyfluoroalkyl substances (PFAS) are a large, diverse group of substances that, in some respects, challenge easy distinction for assessment and management, a clearer understanding of the origin of PFAS found in the environment and assessment of their properties is needed to be able to determine which classes of PFAS require management action. Per- and polyfluoroalkyl substances must be assessed taking into account their differences in chemical, physical, thermal, and biological properties. A single, globally harmonized system for PFAS classification has not yet been defined, resulting in a lack of distinction between PFAS.
A recent review (A Critical Review of the Application of Polymer of Low Concern and Regulatory Criteria to Fluoropolymers, new literature in manuscript,[25]) brings together fluoropolymer toxicity data, human clinical data, and physical, chemical, thermal, and biological data for review and assessment and shows, that fluoropolymers satisfy widely accepted assessment criteria to be considered as “polymers of low concern” (PLC). Details about the distinct rules (based on region) and actual classifications (based on applications) are discussed.
In this regard, we added a short statement and reference to the manuscript (line 115-116, page 3):
Fluoropolymer layers satisfy the widely accepted regulatory assessment criteria to be considered as “polymers of low concern” .[25]
Answer regarding toxicity of precursors, security of the process and scale-up:
C4F8 is a non-toxic gas which has also an E-number (codes for substances used as food additives, E 946). Precursor PFAC-6 is classified as non-toxic and does not contain any hazardous materials with occupational exposure limits established by the region specific regulatory bodies. It is classified as irritating agent (Irritating to eyes, respiratory system and skin). PFAC-8 is classified as not a hazardous substance or mixture. Due to the low flow rate of several µL/min into the process chamber, the concentration of monomer in the chamber is in any case at all times extremely low. Outgassing of the samples does not play a role, because high to complete monomer conversion can be achieved (since non-reacted monomer is constantly converted to reactive parts which polymerize on the substrate or chamber walls). In case of need, excess monomer can be pumped out by the vacuum system, which requires (in case of toxic monomers) specific filters. For the modification of large components (e.g. automotive industry) plasma systems with a chamber volume of up to 8000 L are already in series production (e.g. Tetra-8000 plasma system, Diener electronic GmbH & Co. KG).